# Perceptions of Immersive Virtual Reality for Physical Activity Among Individuals with Hypertension at Risk of Cardiovascular Disease: A Qualitative Study

**DOI:** 10.3390/healthcare13080858

**Published:** 2025-04-09

**Authors:** Jessica García-González, Alberto Verdejo-Herrero, Raúl Romero-del Rey, Héctor García-López, Esteban Obrero-Gaitán, Irene Cortés-Pérez, Raquel Alarcón-Rodríguez

**Affiliations:** 1Department of Nursing, Physical Therapy and Medicine, University of Almeria, 04120 Almeria, Spain; jgg145@ual.es (J.G.-G.); avh275@inlumine.ual.es (A.V.-H.); hector.garcia@ual.es (H.G.-L.); ralarcon@ual.es (R.A.-R.); 2Department of Health Sciences, University of Jaén, 23071 Jaén, Spain; eobrero@ujaen.es (E.O.-G.); icortes@ujaen.es (I.C.-P.)

**Keywords:** exercise, hypertension, obesity, overweight, qualitative research, sedentary behavior, virtual reality

## Abstract

**Background**: Hypertension, obesity, and a sedentary lifestyle are interrelated, forming a vicious cycle that deteriorates cardiovascular health. In addition to being a pathology, hypertension is a risk factor for cardiovascular diseases, one of the leading causes of mortality worldwide. Interventions that combine weight loss and physical activity (PA) reduce cardiovascular risk, but many people face barriers in adhering to regular PA regimens such as a lack of time or motivation. Immersive virtual reality (IVR) has emerged as an innovative alternative to promote PA. This study explored the perceptions of individuals with hypertension and cardiovascular risk regarding the use of IVR as a tool for PA. **Methods**: Fifteen hypertensive adults with cardiovascular risk completed twelve IVR exercise sessions over thirty days. Semi-structured interviews were conducted. **Results**: The thematic analysis identified three main themes: (1) PA, sedentary lifestyle, and health; (2) experiences and perceptions of IVR and PA; and (3) IVR as a useful and safe tool. The participants found IVR engaging, motivating, and effective in overcoming barriers such as a lack of time and social anxiety. Gamification and immersion facilitated greater adherence and enjoyment. **Conclusions**: The participants of this study perceived IVR as an innovative, engaging, and motivating tool for promoting PA. The participants positively valued IVR’s ability to overcome common barriers such as lack of time, adverse weather conditions, and lack of motivation, as well as its immersive and gamified features, which enhanced the adherence to and enjoyment of PA. These results suggest that IVR could complement traditional exercise programs by facilitating the initiation of active routines in sedentary individuals.

## 1. Introduction

The guidelines of the European Society of Cardiology (ESC) and the European Society of Hypertension (ESH) state that blood pressure is considered elevated when it reaches or exceeds 140/90 mmHg when measured in a clinical setting [1]. It is estimated that, globally, 1.28 billion adults between the ages of 30 and 79 suffer from hypertension, with a higher prevalence in low- and middle-income countries. However, about 46% of hypertensive individuals are unaware of their condition, and only 42% of those diagnosed receive treatment [2]. In addition to being a condition in itself, hypertension is a key risk factor for various cardiovascular and kidney diseases, such as ischemic heart disease, heart failure, stroke, chronic kidney disease, and peripheral arterial insufficiency [3]. Its development is influenced by multiple factors, including genetic predisposition, excessive sodium intake, insulin resistance, and obesity [4].

Obesity is a medical condition characterized by an abnormal or excessive accumulation of fat in adipose tissue, reaching levels that can compromise health [5]. According to recent data from the World Health Organization (WHO), in 2022, approximately 2.5 billion adults were overweight, of which more than 890 million were obese (43% and 16% of adults, respectively) [6]. These numbers reflect a significant global health challenge, especially in regions such as Europe, where approximately 25% of adults suffer from obesity, and, considering both overweight and obesity, these conditions affect nearly 60% of the adult population on the continent [7,8]. Long-term data suggest that this global trend of increasing obesity rates has persisted for decades, exacerbating its impact on healthcare systems and economies worldwide. For instance, in 2017, a high body mass index (BMI) was attributed to approximately 4.7 million deaths and 148 million disability-adjusted life years (DALYs) [9], highlighting the severe health consequences of this epidemic.

A sedentary lifestyle is one of the leading contributors to overweight and obesity [10,11]. Sedentary behavior is typically defined as any activity that involves an energy expenditure of fewer than 1.5 metabolic equivalents of task (METs). These activities include sitting or reclining for extended periods, working at a desk, traveling by car or bus, using a computer, or watching television [11,12,13]. In addition to obesity, sedentary behavior is a significant risk factor for cardiovascular diseases [14], the leading cause of mortality globally, accounting for approximately one-third of all deaths [15]. These diseases impose a considerable burden on healthcare systems and economies and if the current trends continue, the economic impact of obesity and overweight is projected to reach USD 4.32 trillion annually by 2035, equivalent to approximately 3% of the global GDP [16].

Hypertension, obesity, and physical inactivity are closely related and form part of a vicious cycle that can severely affect cardiovascular and metabolic health. Treating obesity requires multifaceted approaches, including lifestyle modification, dietary changes, and increased physical activity (PA) [17]. These strategies focus on reducing caloric intake, encouraging balanced nutrition, and promoting PA to increase energy expenditure through structured exercise programs [18]. Pharmacological interventions, such as appetite suppressants and medications that reduce fat absorption, may also be used to support weight loss [19,20]. Additionally, surgical interventions, including adjustable gastric banding, gastric bypass, and sleeve gastrectomy, are available for patients with severe obesity [18].

To mitigate the pathophysiological effects and reduce the cardiovascular risk, interventions that combine weight loss and increased PA have proven effective [14]. However, despite the well-documented benefits of PA, global inactivity levels remain alarmingly high. Approximately 25% of adults and 80% of adolescents worldwide do not meet the recommended physical activity levels, a trend partially driven by societal and lifestyle changes [21]. Many individuals perceive exercise as difficult or boring, particularly after long work or school days, which leads them to prefer sedentary activities such as watching television or playing video games [22].

In response to this challenge, the integration of immersive virtual reality (IVR) with physical activity has emerged as a promising strategy to combat physical inactivity. IVR-based exercise offers an engaging, interactive platform that can transform traditional exercise experiences into more enjoyable and motivating activities. For instance, BOXVR, a popular virtual reality (VR) game, has been shown to meet the World Health Organization’s recommendations for moderate-intensity exercise [23]. Additionally, other applications of IVR have demonstrated benefits in various contexts. These include its use in weight control programs to help adults become more physically active, particularly those prone to overeating after exercise [24]; its ability to enhance enjoyment during high-intensity interval training (HIIT) sessions in individuals with no prior exercise experience, making it an effective method for increasing workout intensity [25]; and the use of virtual reality stationary bikes, which elicit greater physical activity compared to traditional cycling while being perceived as less intense [26]. Furthermore, IVR has demonstrated effectiveness in promoting weight loss, reducing BMI, improving dietary self-control, enhancing adherence to PA programs, and improving body image and self-efficacy in managing health [27,28].

Despite these promising results, research on patient experiences using IVR as an exercise tool, particularly in populations at risk of cardiovascular diseases, remains limited. While preliminary studies suggest positive outcomes, there is a need to understand better how individuals perceive IVR and whether it can be considered a sustainable and effective option for promoting long-term PA engagement [29]. Therefore, the objective of this study was to understand and describe the perceptions of participants with hypertension at risk of cardiovascular disease regarding the use of IVR as a tool for engaging in PA.

## 2. Materials and Methods

### 2.1. Study Design

A descriptive qualitative study was conducted [30]. This type of design allow us to understand and describe experiences, perceptions, and contexts from the perspective of the participants, without excessive interpretation by the researcher. The Consolidated Criteria for Qualitative Research Reporting were used in writing this article [31].

### 2.2. Participants

The participants consisted of 15 subjects from Almería, Spain.

A purposive sampling method was employed [32], using on a database from a private cardiology clinic. Potential participants were identified, contacted, informed about the protocol and study objectives, and invited to participate. The inclusion criteria were as follows: diagnosed with arterial hypertension and undergoing treatment supervised by a cardiologist [33], over 18 years of age, sedentary, overweight or obese (BMI over 25 or 30, respectively) [6], with moderate/high cardiovascular risk, as determined by abdominal circumference according to the WHO (“low risk” ≤ 93 cm in men and ≤79 cm in women; “moderate risk” from 94 to 101 cm in men and from 80 to 87 cm in women; and “high risk” ≥ 102 cm in men and ≥88 cm in women) [34], level 1 (low or inactive) on the International Physical Activity Questionnaire (IPAQ), available to complete the 12 sessions (35 min each) with an IVR device using the BOXVR video game within 30 days, cognitively capable of providing written informed consent, and able to communicate in Spanish or English. The exclusion criteria were that participants could not perform PA due to medical prescription prohibitions or discouraging it due to musculoskeletal system injuries and/or a history of cardiovascular, pulmonary, metabolic diseases, etc. Recruitment was carried out using the WhatsApp instant messaging application for smartphones (Meta Platforms, Inc., Menlo Park, CA, USA) and the snowball concept. Of the 22 participants contacted, 4 declined to participate due to personal reasons and 2 were excluded for regularly engaging in PA. The final sample included 15 participants, as no new information emerged from the interviews, indicating data saturation. Thus, further interviews were deemed unnecessary. The criterion of data saturation was used to determine the necessary number of informants [35].

The 12 sessions were conducted in a room at the University of Almería, with a frequency of 3 sessions per week on alternate days, completed over a total of 30 days. BOXVR is a cardiovascular activity designed to improve functional capacity, although it incorporates movements similar to boxing, which involves a light-to-moderate strength load on the upper limbs [23]. The participants were informed about the safety guidelines related to the exercise [36], and during the sessions, healthcare personnel, including a physiotherapist with experience in cardiac rehabilitation, were always present to ensure the safety and supervision of the activity.

### 2.3. Data Collection

The data collection was completed between June and August 2024 through in-depth semi-structured interviews conducted after completing the final session. The interviews were conducted face-to-face by a study investigator (A.V.-H.) with experience in qualitative research, in a room at the University of Almería. An interview guide, based on a literature review and previously tested with open-ended questions (Table 1), was used. This guide was tested in a pilot group of five participants to ensure that the questions were clear and effective. These questions allowed the participants to provide more detailed responses rather than brief answers. The interviews lasted an average of 30 min, were audio-recorded, transcribed verbatim in Spanish by A.V.-H., and verified by another investigator (R.R.-R.). Before starting the first session, a nurse researcher (J.G.-G.) obtained the sociodemographic data, IPAQ responses, abdominal circumference and body weight measurements [37], as well as informed consent from the participants. The participants were informed that unique identifiers would be used instead of their real names and that the collected data would be stored in secure databases with restricted access. A reflective diary was used to record non-verbal aspects of communication and observations during the interview. These non-verbal aspects were later incorporated into the ATLAS.ti 23 software for further analysis.

### 2.4. Data Analysis

The interview data were analyzed using the qualitative data analysis software ATLAS.TI. A thematic analysis was conducted following the six phases proposed by Kiger and Varpio (2020) [38]: (1) data familiarization, (2) generating initial codes, (3) identifying preliminary themes from data coding, (4) developing and reviewing themes, (5) defining and naming themes, and (6) reporting the findings. After completing the data analysis, the themes and subthemes were established. The coding process was initially performed independently by three researchers (A.V.-H., R.R.-R., and H.G.-L.), who subsequently reached a consensus on the generated codes, meaning units, subthemes, and themes. Team discussions and reviews were used to help resolve discrepancies and strengthen the credibility of the findings. During this process, a table detailing the themes, subthemes, and meaning units was prepared.

### 2.5. Ethical Issues

All the ethical principles established in the Declaration of Helsinki [39] were followed. Each participant was provided with detailed information regarding the study’s purpose, the methodology employed, the voluntary nature of their participation, and their right to withdraw at any time without consequences. Informed consent was obtained from all participants prior to the interviews. Additionally, confidentiality and anonymity were ensured in compliance with Organic Law 3/2018 of 5 December (Data Protection and Guarantee of Digital Rights). Furthermore, the study received approval from the Research Ethics Committee of the Department of Nursing, Physiotherapy, and Medicine at the University.

### 2.6. Rigor and Reliability

The rigor of the study was ensured through the criteria of credibility, confirmability, reliability, and transferability [40]. (1) Credibility: The data were independently analyzed by the researchers. Additionally, the data analysis involved triangulation among the entire research team. Subsequently, the data analysis was reviewed by two researchers experienced in qualitative research. (2) Transferability: The method, study location, participants, and context were described in detail. (3) Reliability: An expert with qualitative research experience, who was not involved in the study, reviewed and confirmed the data analysis. (4) Confirmability: All researchers independently read the transcripts to agree on the emerging meaning units, themes, and subthemes. Finally, the study participants had the opportunity to review and clarify the interpretation of the transcripts.

## 3. Results

### 3.1. Participants Characteristics

A total of 15 participants (10 men and 5 women) aged between 37 and 49 years (mean: 41.93; SD: 4.57) participated in the study. The average weight of the participants was 96.50 kg (SD: 20.8), with an average height of 1.74 m (SD: 0.09) and an average abdominal circumference of 112 cm (SD: 15.9). Five participants had a university education, four had completed high school, and six had secondary education. Thirteen participants had a high cardiovascular risk, of whom, seven were obese (Table 2).

### 3.2. Themes and Subthemes

The qualitative analysis of the data identified three main themes and nine subthemes (Table 3).

### 3.3. Theme 1. PA, Sedentary Lifestyle, and Health

This theme offers a deeper understanding of the participant’s perception of the benefits of PA and the risks associated with a sedentary lifestyle. Additionally, it the analysis of this theme explored the recurring barriers that explain the lack of regular exercise.

#### 3.3.1. Subtheme 1.1. Benefits of PA

Most participants agreed on the importance of exercise as a fundamental tool to promote health and well-being through multiple aspects, and they especially highlighted the benefits related to the cardiovascular system and the prevention of metabolic and chronic diseases.

“*Some of the health benefits of physical exercise that come to mind right now are related to the prevention of cardiovascular diseases and cancer, as well as, of course, the prevention of overweight and obesity…*”.(P-7)

“*It reduces the risk of cardiovascular diseases, blood pressure, colon cancer, diabetes… It helps control overweight and obesity*”.(P-9)

Moreover, the participants emphasized the importance of PA not only in physical terms but also in its emotional and mental well-being and improving sleep quality.

“*I think PA helps prevent mental illnesses like depression, anxiety, helps you sleep well and generally stay fit*”.(P-2)

“*PA improves sleep quality; improves mood because it releases endorphins and makes you feel better; etc.*”(P-3)

#### 3.3.2. Subtheme 1.2. Harms of Sedentary Lifestyle, Overweight, or Obesity

The participants provided a comprehensive perspective on the risks associated with a sedentary lifestyle, highlighting the negative effects on physical health that directly impact daily activities.

“*Not exercising is never positive. If you lead a sedentary life, you are at risk of cardiovascular diseases, increased blood pressure, continuous anxiety, you may have back pain if you are a person like me who sits most of the time in front of a computer for work… you may become obese from not exercising*”.(P-2)

“*Poor lung capacity, muscle pain, joint pain, less agility, in the end, these are small stones that get in the way in your day-to-day routine. At the cardiovascular level… making any effort and not being able to, not being able to take a short walk under fair conditions, not being able to climb stairs, any effort of daily life is harder for you*”.(P-15)

Most interviewees not only highlighted the negative impacts on physical health, including cardiovascular and metabolic diseases, but also emphasized their impact on mental and emotional health.

“*I know this very well because both sedentary lifestyle and obesity have made me feel sad and unmotivated in recent years…*”.(P-3)

“*You don’t feel like doing anything. It affects you mentally, not just physically. It is not ideal to be overweight and sedentary*”.(P-5)

#### 3.3.3. Subtheme 1.3. Reasons for Not Engaging in PA

Among the most mentioned reasons by the interviewees for not engaging in regular PA were the lack of time due to work schedules, family responsibilities, and household chores.

“*I work in an office in the morning. I am always sitting in front of the computer. In the afternoon, the rest of the day, the kids have extracurricular activities like football or catechism, and I have to take care of making the next day’s food. These obligations prevent me from doing daily PA*”.(P-2)

“*Especially taking care of the kids, extracurricular classes, work… That’s what prevents me from exercising*”.(P-9)

Despite knowing the health benefits of PA and the risks of a sedentary lifestyle, some participants pointed to a lack of interest in sports and lack of motivation as barriers to engaging in PA.

“*Going to the gym alone is something I never liked… The activities I could do, like going to the gym, swimming, or cycling, are sports that are mostly done alone, and I don’t like them… Yes, I know I should try to overcome this barrier, but at the moment it is difficult for me*”.(P-5)

“*I don’t like moving, I don’t like exercising. Besides, I am always making excuses, like I lack time. The first thing I eliminate is sports because I don’t like it, and because I haven’t found an activity that completely engages me. I start, but then I get lazy and don’t continue with the routine*”.(P-12)

### 3.4. Theme 2. Experience and Perceptions of IVR and PA

This theme included the interviewees’ previous experiences with IVR before the sessions conducted in this study. Additionally, the analysis of this theme explored the perceptions related to the use of IVR and participation in PA through a device.

#### 3.4.1. Subtheme 2.1. Previous Experience with IVR

Most participants in the study had no prior experience with IVR devices; only three had a previous non-sporting experience.

“*I had never had experience with IVR devices. Well, I did an escape room recently with some glasses, but it told you a story, and you didn’t do anything to interact*”.(P-6)

“*Well, I have only had two VR experiences, one with swords and another musical, playing instruments and drums, with rhythm. They weren’t like this*”.(P-12)

#### 3.4.2. Subtheme 2.2. Perceptions of IVR and PA

One of the aspects highlighted by the participants was the immersive effect that VR generates in the virtual environment where the study sessions took place.

“*I was especially surprised by how it immerses you in that world. It’s amazing how your movements synchronize, reflecting in the game at the speed I was moving*”.(P-13)

“*Being like this intensively, with the music, isolates you from the environment. Throughout the session, it’s just you and the game and nothing else around because you don’t see or hear anything else; you’re immersed there*”.(P-15)

Another notable aspect was the fun and entertainment factor of the activity. The gamification of exercise was highlighted by most interviewees as a strength of IVR and PA.

“*It’s always been said that when something is done through play, you enjoy it more. Practicing exercise as part of a game seems much more motivating than going to a gym to lift weights*”.(P-3)

“*Other forms of exercise are more boring because in a gym, you always do the same thing. Here you can do various types of exercise: faster, slower, moving your feet, moving your hands… I see it more as being inside a game but exercising*”.(P-10)

The participants’ perceptions after exercising with an IVR device were expressed in terms of surprise and enjoyment, influenced by their previous expectations.

“*Honestly, I was very surprised. It activated me like it hadn’t in a long time. I didn’t expect this to be possible*”.(P-4)

“*I thought I would get less tired, like with the Wii, which is always the same. But, my goodness! It feels like I have muscle soreness and everything. It exceeded my expectations*”.(P-12)

### 3.5. Theme 3. IVR as a Useful and Safe Tool

This theme addresses the physical deficits identified by the participants during the sessions, derived from their physical condition, and the analysis of this theme examined the perceptions of IVR as a new tool for exercising.

#### 3.5.1. Subtheme 3.1. Physical Deficits Discovered During IVR Sessions

During their daily routine, most of the participants did not usually experience the repercussions of a physical condition associated with a sedentary lifestyle, overweight, or obesity, mainly due to the lack of more demanding PA than work tasks or responsibilities related to childcare. After exercising with IVR, the participants became aware of their physical limitations.

“*I didn’t know my legs would hurt so much, especially because of the squats. Although I am used to bending down a lot for my two children, I thought I wouldn’t feel so much pain in my legs. My biggest difficulty has been with the squats*”.(P-12)

“*When you face physical demands, you realize that your physical condition is lamentable and think: ’I have to motivate myself to exercise more*”.(P-14)

#### 3.5.2. Subtheme 3.2. A New Way to Exercise

All the participants highlighted IVR’s ability to adapt to different user profiles, without the need to have specific physical conditioning or skills to participate in the activity. Additionally, they mentioned that IVR allows the difficulty levels and demands to be individually adjusted.

“*No specific physical preparation or skill is needed. You can surpass yourself and improve your physical condition gradually. Anyone could do it*”.(P-6)

“*I think anyone could do it, adapting it to their own pace and intensity. As you improve, you can increase speed and improve your coordination according to your level*”.(P-8)

A fundamental pillar to maintaining adherence to PA is intrinsic motivation. The participants expressed a strong motivational drive based on the desire to surpass themselves and progressive improvement from the first to the last session, which was reflected in their greater commitment and enthusiasm during the study sessions.

“*The fact that you feel the need to surpass yourself in each session to try to do much better increases your effort capacity and maintains interest*”.(P-5)

“*Maybe the score, the challenge of surpassing yourself in each session, in one you get a score, and in another… you see that you surpass yourself, that motivates a lot*”.(P-6)

Regarding emotional state, the participants reported feeling better and more active once the sessions were over.

“*I feel good, encouraged. Besides, the music is very energetic, and it makes me want to move, not to stay seated*”.(P-6)

“*I’m tired but fine. Now I’ll take a shower and move on; I’m not going to lie on the couch all afternoon. I feel satisfied and happy*”.(P-12)

#### 3.5.3. Subtheme 3.3. No Excuses to Exercise

Having enough time is a crucial factor in maintaining a regular exercise routine. The time spent traveling to sports centers, the unavailability of equipment due to overcrowding, and other factors represent obstacles for many people. The participants highlighted the option to exercise in a small space and eliminate barriers such as weather conditions and the need to invest large amounts of time.

“*What I liked most was being able to perform the game, exercise in such a small space, and without depending on the weather. In the gym, you often find the machines occupied and have to change plans. Practicing boxing this way is new to me, and I feel comfortable, good after doing it*”.(P-3)

“*The sessions I did this week I liked because they had a contained duration and a realistic objective. They allowed me to adapt them to my available time and physical condition. Probably, I wouldn’t have been able to dedicate two or two and a half hours going to the gym, exercising, showering, and coming back home. Instead, these 30 to 35-min sessions were perfect*”.(P-15)

It was also emphasized that this modality offers a similar experience to group classes without the pressure of feeling observed, which can be beneficial for those with certain inhibitions. It provides an effective exercise experience, similar to a traditional environment like the street or a gym.

“*Most won’t go outside due to shyness or embarrassment, as happens in a gym where others can see you. This way, they can stay at home and keep fit*”.(P-11)

“*It resembles more directed, group classes… but without the worry of feeling ridiculous since you’re alone. Many people are embarrassed to look like a clumsy duck or worry about their physical appearance, and this is an advantage*”.(P-12)

#### 3.5.4. Subtheme 3.4. Recommendations for Inactive People with Cardiovascular Risk

The general view of the interviewees on IVR as a tool for PA focused on the opportunities these devices offer for starting a more active lifestyle and they considered it an excellent way to exercise.

“*If you’re sedentary and do nothing, at least you would be doing some exercise. Burning 500 calories is better than sitting on the couch*”.(P-1)

“*I think for sedentary people who don’t practice any sport, this can be quite beneficial. You don’t need to leave home; simply with your VR glasses, you can exercise. It’s a good substitute, in my opinion*”.(P-5)

## 4. Discussion

This study aimed to understand and describe the perceptions of participants with hypertension and a risk for cardiovascular disease regarding the use of IVR as a tool for PA. The participants highlighted the main benefits of PA and the risks associated with sedentary behavior, emphasizing the effects of PA on the cardiovascular system, emotional health, and sleep quality. They also identified various barriers that hinder regular exercise, such as work schedules and family responsibilities. Additionally, experiences and perceptions regarding IVR in PA were explored, and the results underscored its ability to offer a more entertaining and motivating form of exercise. Finally, the participants described IVR as a useful and safe tool for exercise that is adaptable to different fitness levels. This is particularly relevant given the high prevalence of sedentary behavior and its associated risks, which, as highlighted by Lee, I.M. et al. (2012) [41], are comparable to smoking in terms of mortality risk. The participants believed that IVR could remove common barriers and offered recommendations for inactive people with hypertension at risk of cardiovascular disease, presenting it as a new way to exercise.

Most of the participants emphasized the importance of PA for health, specifically mentioning its benefits in preventing cardiovascular and metabolic diseases. This finding is consistent with the existing literature, which established that regular exercise can significantly reduce the risk of diseases such as hypertension, type 2 diabetes, and certain types of cancer, and helps to control obesity [42,43]. The American Heart Association (AHA) also supports this, stating that regular PA can reduce cardiovascular disease by up to 30% [44]. Additionally, the participants highlighted the benefits of PA on emotional and mental health, noting how regular exercise acts as a natural antidepressant and helps manage disorders such as anxiety [45,46], improves mood, and enhances sleep quality [47], thus contributing to better overall mental health.

The participants also provided a detailed view of the risks of sedentary behavior, highlighting its negative effects on physical and mental health. Research has shown that sedentary behavior increases the risk of mortality from cardiovascular diseases and other health conditions [48]. An unhealthy lifestyle can lead to a reduced lung capacity, muscle and joint pain, and decreased agility, among other issues. These findings align with previous studies that associated sedentary behavior with a higher risk of cardiovascular diseases, obesity, and mental health disorders [49,50,51]. Despite being aware of the benefits of PA and the risks of sedentary behavior, all the participants identified various barriers preventing them from exercising regularly. The most frequently cited reasons included a lack of time, family responsibilities, and a lack of motivation. These difficulties align with the results of other studies that indicated that a lack of interest and motivation for PA is a significant obstacle to adopting a healthier lifestyle, in addition to factors such as work schedules and other environmental conditions [52,53]. In this regard, a recent qualitative systematic review [54] found that hypertensive patients reported time constraints as a persistent barrier to engaging in physical activity. These limitations were primarily related to work commitments, family caregiving responsibilities, and other social obligations, reinforcing the idea that external factors can significantly influence the adoption of a more active lifestyle.

Regarding the use of IVR, most of the participants acknowledged a lack of prior experience with these devices. Additionally, those with previous exposure reported that their experiences were primarily non-sport-related, such as video games or basic interactive applications. This highlights the novelty of this study’s approach in utilizing IVR for PA. Concerning the perceptions of IVR and PA, the participants emphasized two key aspects: gamification and the immersive effect of IVR. Gamification in health has been shown to enhance engagement and adherence to exercise programs [55]. Both factors are essential, as they increase the enjoyment and fun associated with PA, as demonstrated in the study by Seo EY et al. (2023) [56]. In this regard, a recent systematic review and meta-analysis [57] concluded that gamification is an effective tool for promoting participation. Games are inherently appealing, and by incorporating gamified health interventions, it is possible to capture and sustain individuals’ attention, fostering a pleasurable experience and greater engagement. Moreover, gamification facilitates behavior modification, reinforces healthy habits, and motivates users to remain consistently active. Furthermore, the existing literature suggests that PA performed through IVR offers advantages over static exercise and non-immersive VR modalities [58]. In line with these findings, the participants in the present study expressed surprise at the intensity of exercise performed with IVR and its ability to provide a demanding workout. This is consistent with recent research demonstrating that IVR can facilitate moderate-to-vigorous exercise intensities [23,59].

The participants identified various physical deficits they had not previously perceived in their daily routines, revealing a low level of physical fitness. These experiences led them to reflect on the need to improve their fitness to prevent long-term health issues. Additionally, most of the participants found that IVR provided a novel and motivating way to exercise. These findings align with studies highlighting that active IVR games enhance motivation and enjoyment during exercise, significantly improve adaptation to higher training intensities, and consequently lead to better health outcomes [59,60]. Another key aspect valued by the participants was the adaptability of the exercises to different fitness levels and the ability to adjust the intensity. This element of personalization is crucial, as behavior change theories indicate that tailored interventions are associated with better long-term adherence [61]. The ability to overcome personal challenges and receive immediate feedback has also been identified as a key factor in maintaining long-term adherence to physical exercise programs [62,63]. In this regard, a recent study [54] suggested that for individuals lacking the necessary physical activity skills, healthcare professionals could recommend simple exercises and provide practical training using innovative methods such as live videoconferencing and virtual reality to support learning and habit formation. Furthermore, the participants appreciated the ability to exercise at home, in a small space, and without rigid schedules, eliminating common barriers such as a lack of time, adverse weather conditions, and the need to travel to a gym. The possibility of engaging in PA in a safe and comfortable environment is particularly relevant for individuals who have lost confidence in themselves due to body image concerns or fear of social judgment. Various studies support that home-based exercise programs are effective in improving health conditions and serve as a viable alternative to counteract sedentary behavior [64,65]. Finally, the participants recommended IVR as a suitable tool for inactive individuals with cardiovascular risk. They considered IVR an excellent option for initiating an exercise routine and an effective complement to other forms of PA, helping people stay active and reduce sedentary behavior. However, they emphasized the importance of not completely replacing traditional forms of PA, such as outdoor sports, which offer additional benefits, including exposure to nature and social interaction [66].

The findings underline the importance of PA as an essential tool for promoting physical and mental health, highlighting its benefits in preventing cardiovascular, metabolic, and chronic diseases. The participants’ perceptions also highlight the negative effects of sedentary behavior, including the risk of developing diseases and mental health problems. However, a lack of time, motivation, and interest in sports were identified as significant barriers to regular PA. IVR emerged as a promising solution, capable of motivating and maintaining interest through the gamification of exercise, allowing people to exercise effectively at home and adapt to different fitness levels, and eliminating common barriers such as a lack of time and shyness in group settings.

### Strengths and Limitations

The participants exhibit considerable diversity in age, educational level, and life trajectories, providing substantial variability in their responses and enabling a detailed, contextualized understanding of their perceptions and experiences. However, this qualitative study has certain limitations. The findings may have been influenced by the predominant cultural patterns in Western societies and the limited number of participants. Additionally, the study design constrained the generalizability of the results to other populations, does not allow for an assessment of IVR effectiveness in these patients, and relied on the participants’ reported perceptions. Another limitation is the short duration of the intervention (twelve IVR sessions), which may be insufficient to evaluate long-term adherence and health outcomes. Future studies should adopt a mixed-methods approach to validate qualitative findings with quantitative metrics. Furthermore, it would be beneficial to explore IVR interventions in diverse sociocultural contexts to compare and enrich the findings. A longer follow-up period would also help assess sustained engagement and its impact on health [67].

## 5. Conclusions

The findings of this study indicated that the participants perceived IVR as an innovative, engaging, and motivating tool for promoting PA. The participants positively valued IVR’s ability to overcome common barriers such as lack of time, adverse weather conditions, and lack of motivation, as well as its immersive and gamified features, which enhanced their adherence to and enjoyment of PA. These perceptions suggest that IVR could complement traditional exercise programs by facilitating the initiation of active routines in sedentary individuals.

## Figures and Tables

**Table 1 healthcare-13-00858-t001:** Interview guide.

Scenery	Subject	Content/Example Questions
Introduction	My intention	To understand and describe participants’ perceptions after 12 sessions of PA using an IVR device.
	Ethical issues	Inform participants about voluntary participation, recording, consent, confidentiality, and the possibility of withdrawing from the study at any time.
Start	Introductory question	Tell me a little about yourself (Who are you? What do you do?). Have you had previous experience with IRV? How was that experience?
Development	Conversation guide	Do you know the health benefits of PA? Tell me about them. Conversely, how can sedentary behavior and being overweight negatively affect your health? (depending on the response, inquire about cardiovascular risk). As you know, overweight, obesity, and sedentary behavior pose a cardiovascular risk. What reasons prevent or hinder you from regularly engaging in PA? In what ways did the game meet or not meet your expectations as an effective form of exercise? How do you think IVR compares to other forms of exercise you have tried? Do you think the duration of the game sessions is adequate for an effective exercise program? Explain your answer. And do you think IVR can be a useful and safe tool for sedentary, overweight, or obese people? Why?
Closure	Final question	Is there anything else you would like to add?
	Appreciation	Thanks for your participation. Your interview will be incredibly helpful to us.

**Table 2 healthcare-13-00858-t002:** Sociodemographic data.

P	Education Level	Sex	Age	Weight (kg)	Height (m)	BMI	Abdominal Circumference (cm)	CR
1	SE	M	37	87.34	1.76	28.20	103	High
2	SE	F	37	74.60	1.67	26.75	93	High
3	U	M	41	152	1.82	45.89	147	High
4	SE	M	44	98.40	1.67	35.28	126	High
5	HS	M	49	114.20	1.96	29.73	118	High
6	HS	F	43	80.65	1.63	30.35	99	High
7	U	M	46	95.76	1.80	29.56	104	High
8	SE	M	40	85.30	1.76	27.54	101	Moderate
9	SE	F	44	78.70	1.67	28.22	99	High
10	SE	F	48	90.30	1.65	33.17	115	High
11	HS	M	49	120.10	1.78	37.91	131	High
12	U	F	37	74.20	1.63	27.93	87	Moderate
13	U	M	40	101	1.73	33.75	121	High
14	HS	M	37	85	1.71	29.07	109	High
15	U	M	37	110	1.82	33.21	120	High

Abbreviations: HS, high school; SE, secondary education; M, male; F, female; P, participant; CR, cardiovascular risk; U, university.

**Table 3 healthcare-13-00858-t003:** Themes, subthemes, and meaning units.

Theme	Subtheme	Meaning Units
PA, sedentary lifestyle, and health	Benefits of PA	Prevention of cardiovascular diseases, personal well-being, prevention of metabolic disorders, psychological alterations, improved sleep quality, musculoskeletal system improvement, and weight and fat loss
	Harms of Sedentary Lifestyle, Overweight, or Obesity	Cardiovascular problems, depression, anxiety, weight gain, diabetes, fatigue, sleep quality issues, and cancer
	Reasons for not engaging in PA	Lack of time, family responsibilities, and laziness
Experiences and perceptions of IVR and PA	Previous experience with IVR	No previous experience, non-sporting experiences
	Perceptions of IVR and PA	Entertaining, realistic, immersion, isolation from reality, concentration, surprising, and demanding
IVR as a useful and safe tool	Physical deficits discovered during IVR sessions	Poor physical condition, muscle pain, and limited reaction capacity
	A new way to exercise	Adaptable, fatigue, mood improvement, sweating, exercise adherence, motivation, and self-improvement
	No excuses for exercising	Does not require much time investment, no schedules, at-home activity, no weather impediments, and convenience
	Recommendations for inactive people with cardiovascular risk	New experience, good way to start, good sports complement, getting out of a sedentary lifestyle, and embarrassment

## Data Availability

For confidentiality purposes, the data are in the possession of the author (R.R.-d.R.).

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
