# Peer review of "Perceptions of Immersive Virtual Reality for Physical Activity Among Individuals with Hypertension at Risk of Cardiovascular Disease: A Qualitative Study"

_healthcare, 2025, doi:10.3390/healthcare13080858_

Round 1

Reviewer 1 Report

Comments and Suggestions for Authors

This study explores the perceptions of individuals with hypertension and cardiovascular risk regarding the use of immersive virtual reality (IVR) for physical activity. While the topic is relevant and well-motivated in the introduction, serious methodological limitations undermine the scientific validity of the findings.
The most critical flaw of this study is its extremely small sample size of only 15 participants. Such a limited group makes it impossible to generalize the results to a broader population. Moreover, the non-probabilistic selection method raises concerns about selection bias, further reducing the representativeness of the sample. As a result, the findings cannot be considered reliable or applicable in a clinical or public health context.
The methodological approach also presents significant weaknesses. The authors use thematic qualitative analysis, which is suitable for exploratory studies, but given the small sample size, it does not provide a strong basis for drawing meaningful conclusions. Additionally, the lack of a control group makes it difficult to assess the actual effect of IVR on participants’ physical activity levels.
While the results are clearly presented, their analytical value remains low. The authors fail to compare their findings to larger-scale studies that could either validate or contradict their observations. Without such comparisons, it is impossible to assess whether the observed effects of IVR are significant or merely coincidental.
The discussion section highlights the potential benefits of IVR, but given the weak empirical foundation of this study, these conclusions remain speculative. The authors do not sufficiently acknowledge the limitations of their work, suggesting a lack of critical engagement with their own findings.
In conclusion, this article does not meet the methodological standards required for a scientific study that can serve as a basis for further research or practical applications. Due to the insufficient sample size and the lack of a rigorous comparative analysis, the conclusions presented are unreliable and should not inform any policy or clinical recommendations regarding IVR for hypertension and cardiovascular risk.
I recommend rejecting the article and suggesting that the authors conduct a study with a larger, more diverse sample, incorporating a control group and quantitative analysis methods to improve the reliability and validity of their findings.

Author Response

Dear Reviewer 1,

This study explores the perceptions of individuals with hypertension and cardiovascular risk regarding the use of immersive virtual reality (IVR) for physical activity. While the topic is relevant and well-motivated in the introduction, serious methodological limitations undermine the scientific validity of the findings.

Comments 1: The most critical flaw of this study is its extremely small sample size of only 15 participants. Such a limited group makes it impossible to generalize the results to a broader population. Moreover, the non-probabilistic selection method raises concerns about selection bias, further reducing the representativeness of the sample. As a result, the findings cannot be considered reliable or applicable in a clinical or public health context.

Response 1: We thank the reviewer for their comments and for the time dedicated to evaluating our study. We understand their concern regarding the sample size and participant selection; however, we believe that these aspects should be analyzed from the qualitative paradigm in which our research is framed. In qualitative studies, the emphasis is not on statistical generalization but on the depth and richness of the information obtained.

Unlike quantitative studies, sample size in qualitative research is not determined a priori but is established based on the criterion of data saturation—when interviews no longer provide new relevant information. In this regard, the number of participants in our study was adequate to meet this criterion.

As with all qualitative studies, our research does not aim to extrapolate results to a broader population but rather to gain an in-depth understanding of participants' lived experiences within their sociocultural context.

References:

Saunders, B., Sim, J., Kingstone, T., Baker, S., Waterfield, J., Bartlam, B., Burroughs, H., & Jinks, C. (2018). Saturation in qualitative research: exploring its conceptualization and operationalization. Quality & Quantity, 52(4), 1893–1907. https://doi.org/10.1007/S11135-017-0574-8

Ahmad, M., Wilkins, S. Purposive sampling in qualitative research: a framework for the entire journey. Qual Quant (2024). https://doi.org/10.1007/s11135-024-02022-5

Comments 2: The methodological approach also presents significant weaknesses. The authors use thematic qualitative analysis, which is suitable for exploratory studies, but given the small sample size, it does not provide a strong basis for drawing meaningful conclusions. Additionally, the lack of a control group makes it difficult to assess the actual effect of IVR on participants’ physical activity levels.

Response 2: Thank you for your comments. We would like to clarify that the objective of our study was not to evaluate the effect of IVR on physical activity levels. To achieve this, an experimental design, such as a randomized controlled trial, would indeed have been necessary. However, our study follows a qualitative approach aimed at exploring the perceptions and experiences of individuals with hypertension and cardiovascular risk regarding the use of IVR as a tool for physical activity.

In this type of study, the use of a control group is not relevant, as the approach does not seek to establish causality or make quantifiable comparisons but rather to gain an in-depth understanding of participants' lived experiences. The thematic qualitative analysis employed is a widely accepted methodology for this type of research and is appropriate for addressing our research question.

We consider that the methodological quality of the study is suitable for the stated objectives, and we trust that this clarification helps to contextualize the study’s approach within the qualitative paradigm. We have attached the references supporting the chosen methodological approach.

References:

Colorafi, K.J.; Evans, B. Qualitative Descriptive Methods in Health Science Research. HERD 2016, 9, 16–25

Kiger, M. E., & Varpio, L. (2020). Thematic analysis of qualitative data: AMEE Guide No. 131. Medical Teacher, 42(8), 846–854. https://doi.org/10.1080/0142159X.2020.1755030

Comments 3: While the results are clearly presented, their analytical value remains low. The authors fail to compare their findings to larger-scale studies that could either validate or contradict their observations. Without such comparisons, it is impossible to assess whether the observed effects of IVR are significant or merely coincidental.

Response 3: Thank you for your comment. Since our study employs a qualitative approach, its objective is not to determine the statistical significance of IVR effects but rather to explore participants' experiences and perceptions. Given that our goal is to explore and describe these aspects, the results do not include quantitative analyses or statistical inference. We have included a suggestion for future studies to incorporate a mixed-methods approach that strengthens qualitative findings with quantitative measurements.

Comments 4: The discussion section highlights the potential benefits of IVR, but given the weak empirical foundation of this study, these conclusions remain speculative. The authors do not sufficiently acknowledge the limitations of their work, suggesting a lack of critical engagement with their own findings.

Response 4: We appreciate your comments and recognize the importance of critically reflecting on the limitations of our study. While our discussion highlights the potential benefits of IVR, we acknowledge that the empirical evidence is still emerging and that our findings should be interpreted with caution.

To address this concern, we have revised and expanded the discussion and limitations section of the study. Additionally, we have reformulated the conclusions to focus on participants' perceptions of IVR as a tool for physical activity, avoiding statements regarding its safety and effectiveness—as these aspects were not directly assessed in our research.

We sincerely appreciate this observation, as it has allowed us to enhance the rigor of our analysis and the clarity of our work.

Comments 5: In conclusion, this article does not meet the methodological standards required for a scientific study that can serve as a basis for further research or practical applications. Due to the insufficient sample size and the lack of a rigorous comparative analysis, the conclusions presented are unreliable and should not inform any policy or clinical recommendations regarding IVR for hypertension and cardiovascular risk.

I recommend rejecting the article and suggesting that the authors conduct a study with a larger, more diverse sample, incorporating a control group and quantitative analysis methods to improve the reliability and validity of their findings.

Response 5: We sincerely appreciate the reviewer's comments and the time dedicated to evaluating our study. However, we would like to point out that this is a qualitative study and, as such, should be assessed within the methodological framework of this approach rather than from a quantitative perspective. In this regard, we believe that some comments do not reflect inherent weaknesses in the manuscript but may stem from an interpretation based on quantitative principles and methods that are not applicable to this type of research.

Qualitative research does not aim to generalize results or establish causal relationships in the same way as quantitative studies. Instead, it seeks to explore participants' perceptions and experiences in depth, which is reflected in our methodology and study design. Furthermore, in qualitative research, sample size is determined based on data saturation rather than the need for statistical representativeness.

We understand that the reviewer may have expected a quantitative approach, but we believe this is not a valid critique for a study of qualitative nature. We appreciate the suggestion to expand the sample and incorporate quantitative methods; however, given the exploratory objective of our study, a qualitative design is the most appropriate to address our research question.

We hope this clarification helps contextualize our research within its methodological paradigm, and we once again appreciate the opportunity to address the reviewer's comments.

Reviewer 2 Report

Comments and Suggestions for Authors

Congratulations on your work. I don't have many suggestions to make, just check the reference on line 388 "Lee, I., et al. (2012)[30]". Is it in accordance with the guidelines?

Additional comments:

- The present article on the inclusion of virtual reality as a strategy to increase physical activity levels has a very interesting practical intervention component.
- The selected questions and research analysis topics seem to be well aligned with the proposed research objectives.
- Methodologically, the study is well-structured.
- The results presented are in accordance with the stated objectives, and the discussion addresses both the importance of physical activity for health and the barriers commonly reported by individuals.
- The conclusions present the ideas obtained in this work clearly.   As a proposal for future work, they could include quantitative data on blood pressure before and after the intervention.

Best regards.

Author Response

Dear Reviewer 2,

Comments 1: Congratulations on your work. I don't have many suggestions to make, just check the reference on line 388 "Lee, I., et al. (2012)[30]". Is it in accordance with the guidelines?

Response 1: We appreciate the comment and the observation regarding the typographical error. We have corrected the reference, which now appears as follows: Lee, I.M., et al. (2012).

Reviewer 3 Report

Comments and Suggestions for Authors

The aim of the study was to investigate the perception of people with hypertension and cardiovascular risk in relation to the use of IVR as a tool for physical activity. The manuscript addresses the current issue of a sedentary lifestyle in the context of civilization diseases and a possible way to increase physical activity through the use of modern technological possibilities. 

In order to improve the quality of the manuscript, it is necessary to supplement the methodological part with an appropriate description and justification of the chosen research methodology (why was the qualitative study chosen?) and to deepen the description of the methods of analyzing data from the qualitative study. The manuscript requires reformulation of conclusions. Detailed instructions can be found in the appendix.

Reviewer's comments:

Article:
Perceptions of Immersive Virtual Reality for Physical Activity among Individuals with Hypertension and Cardiovascular Risk: A Qualitative Study.

Introduction

Q1: It is worth supplementing the introduction with more examples of IRV applications in the context of increasing physical activity (Spanish, European, global).

Materials and Methods

Participants:

Q2: Participant Recruitment: Details of the recruitment process are insufficient. The manuscript should explain the approach used (e.g., purposive sampling, convenience sampling) and cite appropriate methodological references. Where did the database of numbers for sending invitations come from? Who were the people to whom the text messages were sent: e.g., clinic patients?

Q3: Who and how performed circumferential measurements, body weight, diagnosed hypertension in patients and on what basis (which was an inclusion criterion)?

Q4: Were these measurements taken before and at the end of the study?

Q5: Were patients informed about safety rules regarding exercise, e.g. what blood pressure values ​​constitute a contraindication to starting exercise?

Q6: Were the study participants' hypertension treated and monitored by a physician?

Data collection:

Q7: How many people participated in the pilot study?

Q8: What measures are in place to ensure secure storage of interview data?

Q9: What was the nature of the exercises that the participants performed in the study? – were they mostly strength exercises, cardio?

Q10: How do we know that the study participants did the recommended amount of exercise during the study?

Q11: Were the patients informed about the exercise plan, i.e. were they supposed to do the recommended amount of exercise every other day, or could it be that the patient did 12 sessions in a row and did nothing the other days of the month?

Data Analysis

Q12: Interview Analysis: The manuscript lacks detail on how the qualitative data were analyzed. A clear description of the coding framework or thematic analysis method is required, and the appropriate literature supporting the chosen method.

Strengths and Limitations

Q13: In the limitations section, it is worth mentioning the small number of study participants.

Discussion and conclusions

Q14: The discussion requires stronger connections to prior research. Linking findings to established literature would enhance the manuscript’s depth.

Q15: I would avoid writing about financial benefits (reduction in healthcare costs) without specific analyses and examples, especially if this was not the aim or subject of the study.

Q16: The conclusion includes the sentence: „IVR is an effective and safe tool for initiating and maintaining a PA routine in individuals with hypertension and cardiovascular risk”

Can we definitely say this based on the research conducted? On what basis and in what way was the safety of the intervention and its effectiveness (measurement parameters) assessed, and did the study involve enough participants to be able to draw such a conclusion? Perhaps it would be more appropriate to state that: in the opinion of the study participants, IRV is an attractive form of physical activity that counteracts a sedentary lifestyle. In order to avoid drawing too broad conclusions. The aim of the study was to examine the perception of users regarding the use of IRV and this is what should be referred to primarily when drawing conclusions. The remaining aspects were not studied.

Author Response

Dear Reviewer 3,

INTRODUCTION

Comments 1: It is worth supplementing the introduction with more examples of IRV applications in the context of increasing physical activity (Spanish, European, global).

Response 1: We appreciate the reviewer's valuable suggestion, which will undoubtedly enhance our introduction. In response to this recommendation, we have incorporated several studies that explore IVR applications in promoting physical activity. These additional references provide a more comprehensive perspective on the impact of IVR in different contexts and populations, further reinforcing the relevance of the topic addressed in our study. Below, we present the added information:

Additionally, other applications of IVR have demonstrated benefits in various contexts. These include: its use in weight control programs to help adults become more physically active, particularly those prone to overeating after exercise; its ability to enhance enjoyment during high-intensity interval training (HIIT) sessions in individuals with no prior exercise experience, making it an effective method for increasing workout intensity; and the use of virtual reality stationary bikes, which elicit greater physical activity compared to traditional cycling while being perceived as less intense.

References:

  • Sauchelli, S.; Brunstrom, J.M. Virtual Reality Exergaming Improves Affect during Physical Activity and Reduces Subsequent Food Consumption in Inactive Adults. Appetite 2022, 175, doi:10.1016/J.APPET.2022.106058.
  • Farrow, M.; Lutteroth, C.; Rouse, P.C.; Bilzon, J.L.J. Virtual-Reality Exergaming Improves Performance during High-Intensity Interval Training. Eur J Sport Sci 2019, 19, 719–727, doi:10.1080/17461391.2018.1542459.
  • Zeng, N.; Liu, W.; Pope, Z.C.; McDonough, D.J.; Gao, Z. Acute Effects of Virtual Reality Exercise Biking on College Students’ Physical Responses. Res Q Exerc Sport 2022, 93, 633–639, doi:10.1080/02701367.2021.1891188.

MATERIALS AND METHODS

Participants:

Comments 2: Participant Recruitment: Details of the recruitment process are insufficient. The manuscript should explain the approach used (e.g., purposive sampling, convenience sampling) and cite appropriate methodological references. Where did the database of numbers for sending invitations come from? Who were the people to whom the text messages were sent: e.g., clinic patients?

Response 2: We appreciate the reviewer's comments regarding the recruitment process. In response to this observation, we have expanded the relevant section (2.2. Participants) to provide more details on the methodological approach used, specifying that purposive sampling was employed. Additionally, we have included an appropriate methodological reference and clarified the source of the database used for sending invitations, as well as the characteristics of the recipients of the text messages. These modifications add greater clarity and rigor to the process described in the manuscript.

Reference:

  • Ahmad, M., Wilkins, S. Purposive sampling in qualitative research: a framework for the entire journey. Qual Quant (2024). https://doi.org/10.1007/s11135-024-02022-5

Comments 3: Who and how performed circumferential measurements, body weight, diagnosed hypertension in patients and on what basis (which was an inclusion criterion)?

Response 3: Thank you for your comments. The information has been added to clarify that abdominal circumference and body weight measurements were taken by a nurse researcher, following previously established protocols. Similarly, a reference has been included to clarify the diagnostic criteria for hypertension used by the cardiologist.

References:

  • Ross, R., Neeland, I.J., Yamashita, S. et al. Waist circumference as a vital sign in clinical practice: a Consensus Statement from the IAS and ICCR Working Group on Visceral Obesity. Nat Rev Endocrinol 16, 177–189 (2020). https://doi.org/10.1038/s41574-019-0310-7
  • John William McEvoy, Cian P McCarthy, Rosa Maria Bruno, Sofie Brouwers, Michelle D Canavan, Claudio Ceconi, Ruxandra Maria Christodorescu, Stella S Daskalopoulou, Charles J Ferro, Eva Gerdts, Henner Hanssen, Julie Harris, Lucas Lauder, Richard J McManus, Gerard J Molloy, Kazem Rahimi, Vera Regitz-Zagrosek, Gian Paolo Rossi, Else Charlotte Sandset, Bart Scheenaerts, Jan A Staessen, Izabella Uchmanowicz, Maurizio Volterrani, Rhian M Touyz, ESC Scientific Document Group , 2024 ESC Guidelines for the management of elevated blood pressure and hypertension: Developed by the task force on the management of elevated blood pressure and hypertension of the European Society of Cardiology (ESC) and endorsed by the European Society of Endocrinology (ESE) and the European Stroke Organisation (ESO), European Heart Journal, Volume 45, Issue 38, 7 October 2024, Pages 3912–4018, https://doi.org/10.1093/eurheartj/ehae178

Comments 4: Were these measurements taken before and at the end of the study?

Response 4: Thank you for your comment. The measurements were taken prior to the initiation of the study sessions. The purpose of these measurements was to include participants in the study and to describe the characteristics of our interviewees. Since this was a qualitative study, no measurements were taken at the end, as no data comparison was intended, as would be the case in quantitative studies. A mention of this has been included in the study's limitations, and we have suggested the incorporation of mixed methodologies for future studies.

Comments 5: Were patients informed about safety rules regarding exercise, e.g. what blood pressure values ​​constitute a contraindication to starting exercise?

Response 5: Thank you for your comment. Participants were informed about safety guidelines, including contraindications related to blood pressure. Additionally, physical activity through IVR was carried out in the presence of a physiotherapist specialized in cardiac rehabilitation. To enhance clarity, this information has been added to section 2.2. Participants.

Reference:

  • Sharman, J. E., Smart, N. A., Coombes, J. S., & Stowasser, M. (2019). Exercise and sport science australia position stand update on exercise and hypertension. Journal of human hypertension, 33(12), 837–843. https://doi.org/10.1038/s41371-019-0266-z

Comments 6: Were the study participants' hypertension treated and monitored by a physician?

Response 6: Thank you for your observation. All participants in the study had a diagnosis of arterial hypertension and were receiving treatment supervised by a cardiologist at a private clinic in Almería, Spain. For greater clarity, we have incorporated this information into the inclusion criteria: Diagnosed with arterial hypertension and undergoing treatment supervised by a cardiologist.

Data collection:

Comments 7: How many people participated in the pilot study?

Response 7: Thank you for your comment. The question guide was tested with a pilot group of five participants to ensure the clarity and effectiveness of the questions to be included in the interviews. This information has been added to the methodology section (2.3. Data collection).

Comments 8: What measures are in place to ensure secure storage of interview data?

Response 8: We appreciate the reviewer's comment, as it has allowed us to identify an aspect that was not previously reflected in the manuscript. To address this omission, we have added the following information in the methodology section (2.3. Data collection), specifying the measures implemented to ensure the secure storage of interview data.

Participants were informed that unique identifiers would be used instead of their real names and that the collected data would be stored in secure databases with restricted access.

Comments 9: What was the nature of the exercises that the participants performed in the study? – were they mostly strength exercises, cardio?

Response 9: Thank you for these questions. We have added the relevant information to clarify this aspect (2.2. Participants). The nature of the exercise was primarily cardiovascular, aimed at improving functional capacity. However, during its execution, movements similar to those of boxing were performed, which involved a light to moderate load of strength in the upper limbs.

Comments 10: How do we know that the study participants did the recommended amount of exercise during the study?

Response 10: Thank you for your question. Each IVR session was supervised by a physiotherapist, who was responsible for ensuring that participants performed the recommended amount of exercise. We have added this information in a new paragraph in section 2.2. Participants to clarify this aspect.

Comments 11: Were the patients informed about the exercise plan, i.e. were they supposed to do the recommended amount of exercise every other day, or could it be that the patient did 12 sessions in a row and did nothing the other days of the month?

Response 11: Thank you for your question. In line with the previous issue, a new paragraph has been added in section 2.2. Participants to clarify this aspect. Participants were scheduled three times a week on alternate days over 4 weeks, allowing them to complete a total of 12 sessions within a month. This ensured that the amount of exercise was performed according to the established plan, and participants were not allowed to complete all sessions consecutively.

Data Analysis

Comments 12: Interview Analysis: The manuscript lacks detail on how the qualitative data were analyzed. A clear description of the coding framework or thematic analysis method is required, and the appropriate literature supporting the chosen method.

Response 12: We appreciate the reviewer's comment. A thematic analysis was conducted following the methodology proposed by Kiger and Varpio (2020), which provides a structured approach for identifying, analyzing, and interpreting patterns in qualitative data. To improve the clarity and transparency of the analytical process, we have expanded subsection 2.5 (Data analysis), detailing the phases of the analysis, including initial coding, identification of units of meaning, generation of subthemes and themes, as well as the consensus process among the researchers. Additionally, we have incorporated an additional reference that supports our methodological choice. Below, we present the added information:

Thematic analysis was conducted following the six phases proposed by Kiger and Varpio (2020): (1) data familiarization, (2) generating initial codes, (3) identifying preliminary themes from data coding, (4) developing and reviewing themes, (5) defining and naming themes, and (6) reporting the findings. After completing the data analysis, the themes and subthemes were established. he coding process was initially performed independently by three researchers (A.V.-H., R.R.-R., and H.G.-L.), who subsequently reached a consensus on the generated codes, meaning units, subthemes, and themes.

Reference: 

  • Kiger, M. E., & Varpio, L. (2020). Thematic analysis of qualitative data: AMEE Guide No. 131. Medical Teacher, 42(8), 846–854. https://doi.org/10.1080/0142159X.2020.1755030

Strengths and Limitations

Comments 13: In the limitations section, it is worth mentioning the small number of study participants.

Response 13: We appreciate the reviewer's comment and have added additional information to the limitations section of the study regarding the number of participants.

Discussion and conclusions

Comments 14: The discussion requires stronger connections to prior research. Linking findings to established literature would enhance the manuscript’s depth.

Response 14: We sincerely appreciate your valuable suggestion. To strengthen the discussion, we have expanded the analysis of our findings in relation to previous studies, establishing stronger links with the existing literature. Additional information has been incorporated regarding the impact of gamification and exercise personalization on motivation and adherence to physical activity. These studies reinforce the importance of interventions adapted to fitness levels and the potential of IVR to overcome the barriers identified by our participants. The references to these works have been included to enrich the discussion and provide a more robust framework on how emerging technologies can positively influence the promotion of an active lifestyle. We greatly appreciate your comments, which have been instrumental in improving the depth and quality of the manuscript.

References:

  • Alzghoul B. The Effectiveness of Gamification in Changing Health-related Behaviors: A Systematic Review and Meta-analysis. Open Public Health J, 2024; 17: e18749445234806. http://dx.doi.org/10.2174/0118749445234806240206094335
  • Li, Q., Jiang, J., Duan, A. et al. Physical activity experience of patients with hypertension: a systematic review and synthesis of qualitative literature. BMC Public Health 24, 2826 (2024). https://doi.org/10.1186/s12889-024-20326-x

Comments 15: I would avoid writing about financial benefits (reduction in healthcare costs) without specific analyses and examples, especially if this was not the aim or subject of the study.

Response 15: We appreciate the reviewer's suggestion. Since our study did not aim to analyze the economic benefits, we have removed such statements from the manuscript to avoid unfounded interpretations.

Comments 16: The conclusion includes the sentence: „IVR is an effective and safe tool for initiating and maintaining a PA routine in individuals with hypertension and cardiovascular risk”

Can we definitely say this based on the research conducted? On what basis and in what way was the safety of the intervention and its effectiveness (measurement parameters) assessed, and did the study involve enough participants to be able to draw such a conclusion? Perhaps it would be more appropriate to state that: in the opinion of the study participants, IRV is an attractive form of physical activity that counteracts a sedentary lifestyle. In order to avoid drawing too broad conclusions. The aim of the study was to examine the perception of users regarding the use of IRV and this is what should be referred to primarily when drawing conclusions. The remaining aspects were not studied.

Response 16: We appreciate the reviewer's observation and agree that the conclusion should align with the study's objective. Therefore, we have reworded the conclusions to focus on the participants' perception of IVR as a tool for physical activity, avoiding statements about its safety and effectiveness, aspects that were not directly assessed in our research.

Conclusions: 

The findings of this study indicated that participants perceived IVR as an innovative, engaging, and motivating tool for promoting PA in individuals with hypertension and cardiovascular risk. Participants positively valued IVR's ability to overcome common barriers such as lack of time, adverse weather conditions, and lack of motivation, as well as its immersive and gamified features, which enhanced adherence and enjoyment of PA. These perceptions suggest that IVR could complement traditional exercise programs by facilitating the initiation of active routines in sedentary individuals.

Round 2

Reviewer 1 Report

Comments and Suggestions for Authors

Thank you to the authors for responding to the review. I recommend publication of the manuscript.

Reviewer 3 Report

Comments and Suggestions for Authors

The changes made the manuscript more coherent and clear to the reader.